# Comprehensive Linear Epitope Prediction System for Host Specificity in *Nodaviridae*

**DOI:** 10.3390/v14071357

**Published:** 2022-06-22

**Authors:** Tao-Chuan Shih, Li-Ping Ho, Hsin-Yiu Chou, Jen-Leih Wu, Tun-Wen Pai

**Affiliations:** 1Department of Computer Science and Information Engineering, National Taipei University of Technology, Taipei 10608, Taiwan; t108599001@ntut.org.tw; 2Department of Aquaculture, National Penghu University of Science and Technology, Penghu 88046, Taiwan; lphotw@gms.npu.edu.tw; 3Department of Aquaculture, College of Life Science, National Taiwan Ocean University, Keelung 20224, Taiwan; hychou@mail.ntou.edu.tw; 4Department of Bioscience and Biotechnology, National Taiwan Ocean University, Keelung 20224, Taiwan; jlwu@gate.sinica.edu.tw; 5Institute of Cellular and Organismic Biology, Academia Sinica, Taipei 11529, Taiwan

**Keywords:** linear epitope *Nodaviridae*, host specificity, multi-expert prediction

## Abstract

Background: *Nodaviridae* infection is one of the leading causes of death in commercial fish. Although many vaccines against this virus family have been developed, their efficacies are relatively low. *Nodaviridae* are categorized into three subfamilies: alphanodavirus (infects insects), betanodavirus (infects fish), and gammanodavirus (infects prawns). These three subfamilies possess host-specific characteristics that could be used to identify effective linear epitopes (LEs). Methodology: A multi-expert system using five existing LE prediction servers was established to obtain initial LE candidates. Based on the different clustered pathogen groups, both conserved and exclusive LEs among the *Nodaviridae* family could be identified. The advantages of undocumented cross infection among the different host species for the *Nodaviridae* family were applied to re-evaluate the impact of LE prediction. The surface structural characteristics of the identified conserved and unique LEs were confirmed through 3D structural analysis, and concepts of surface patches to analyze the spatial characteristics and physicochemical propensities of the predicted segments were proposed. In addition, an intelligent classifier based on the Immune Epitope Database (IEDB) dataset was utilized to review the predicted segments, and enzyme-linked immunosorbent assays (ELISAs) were performed to identify host-specific LEs. Principal findings: We predicted 29 LEs for *Nodaviridae*. The analysis of the surface patches showed common tendencies regarding shape, curvedness, and PH features for the predicted LEs. Among them, five predicted exclusive LEs for fish species were selected and synthesized, and the corresponding ELISAs for antigenic feature analysis were examined. Conclusion: Five identified LEs possessed antigenicity and host specificity for grouper fish. We demonstrate that the proposed method provides an effective approach for in silico LE prediction prior to vaccine development and is especially powerful for analyzing antigen sequences with exclusive features among clustered antigen groups.

## 1. Introduction

*Nodaviridae* infection is a common cause of death in marine animals and insects, and the virus family is classified into several genera according to host specificity. To date, various vaccines have been developed for aquaculture, including recombinant proteins, synthetic peptides, inactivated virions, DNA vaccines, and virus-like particles. However, the efficacy of these vaccines remains unsatisfactory. Therefore, a more effective immunization strategy and a comprehensive vaccine development against these viruses are important for maintaining commercially viable fisheries. Since *Nodaviridae* show a wide host range of fishes and prawns and the mechanism is controlled by major capsid protein (MCP), we proposed a multi-expert voting mechanism, host-specific, and surface structural analytics of *Nodaviridae* linear epitopes (LEs) for each subfamily.

*Nodaviridae* are a family of non-enveloped RNA viruses that contain two major infectious segments: RNA1 and RNA2. In addition, the subgenome RNA3 is produced from RNA1 during RNA replication. RNA1 (3.2 kb) encodes the RNA-dependent RNA polymerase (RdRp) and non-structural B2 like protein and functions by selecting appropriate RNA templates and initiation sites for RNA replication. RNA2 (1.2 kb) encodes a viral capsid protein (CP) and plays an important role in inducing apoptosis, followed by secondary necrosis of infected cells through the mitochondria-mediated cell death pathway [1]. In addition, the CP of *Nodaviridae* assembles into viral particles with icosahedral structures. The CP is composed of core jelly-roll topology, forming a face-to-face beta sandwich with two pairs of anti-parallel beta sheets [2]. CPs from specific *Nodaviridae* genera are categorized into three major domains. The first domain is an N-terminal arginine-rich region (N-ARM) contributed to the formation of a CP via hydrogen bonding and interacts with the RNA genome. The second shell domain (S-domain) consists of 60 trimeric S-domains that participate in inter-subunit contacts and forms a continuous thin shell with an empty inner cavity as an icosahedral structure. The third protrusion domain (P-domain) forms a protrusion structure on the surface particle and is characterized by poor electron density with high flexibility. Previous reports have validated residues 247 and 270 as host-binding sites for turbot during viral infection [2,3,4,5,6,7]. Sub-genomic RNA3 encodes one or two nonstructural B1 and B2 proteins. These segments help repress antiviral responses in Nodamura-infected cells and play a role in the inhibition of host RNA interference.

The family *Nodaviridae* belongs to the *Riboviria* realm, phylum incertae sedia. According to the ICTV’s virus taxonomy 2018b [8], the family *Nodaviridae* is categorized into three genera. The first is alphanodaviridae, which exhibit the broadest range of host specificity for invertebrate species. A cleave mechanism to yield infectivity particles exists in all known insect nodaviruses and the cleavage site is conserved within all insect-infect viruses [9]. The other genera of the betanodavirus subfamily only infect fish species. Betanodaviruses are known to affect over 120 fish species, particularly groupers and seabass [4]. The mortality of infected hosts at the larval stage has reaches of 100% [2,10]. Apart from horizontal transmission, betanodaviruses can be transmitted vertically through gonad infections, with parents passing the virus to their progenies [4]. Compared with alphanodaviruses, betanodaviruses exhibit a less conserved cleavage site for autocatalytic proteolysis and low host specificity. Betanodaviruses can be classified into four genera with highly variable RNA2 sequences: *Tiger puffer nervous necrosis virus* (TPNNV), *Redspotted*
*grouper nervous necrosis virus* (RGNNV), *Barfin flounder nervous necrosis virus* (BFNNV), and *Striped jack nervous necrosis virus* (SJNNV). The different genotypes are correlated in their host range. For example, RGNNV exhibits the widest host range for warm-water fish species. In contrast, BFNNV is known to infect cold-water fish species. TPNNV infects only tiger pufferfish. SJNNV was initially restricted to Japanese waters, however it was also found in southern European waters. Although *betanodaviridae* viruses show high sequence similarity (>80%) with the coat protein sequence, reassortant viruses carrying *Sevenband grouper nervous necrosis virus* (SGNNV) are unable to infect striped jacks. Conversely, reassortant viruses with SJNNV cannot cause an infection in sevenband groupers [11]. These infectious events and genotypes tend to be associated with specific water temperatures (15–20 °C for BFNNV, 20 °C for TPNNV, and 20–25 °C for SJNNV), which reflects the geographic distribution of distinct fish species. The attachment of nervous necrosis to stress and heat shock–induced heat shock cognate protein (Hsc70) plays an important role in infection. The antigenicity of NNV particles is known to change due to the disruption of their conformational structures [5,11]. Recent reports have suggested that prawn-infecting species, including MrNV and PvNV nodaviruses, should be categorized into a new gammanodavirus genus due to their distinct genomic sequences compared with alphanodaviruses and betanodaviruses. Although MrNV does not cause death in adult prawns, they might serve as virus carriers to transmit viruses and are known to cause 100% mortality in larval and post-larval prawns [4].

Structural similarity analysis of NNV revealed that there was no significant homology in CP sequences and structures among the three *Nodaviridae* genera. However, these *Nodaviridae* did share a similar composition. Three neighboring monomeric S-domains with P-domain comprise 8 anti-parallel beta strands, which construct a twisted sheet or a jelly-roll fold, from subunits A, B, and C assembled into trimeric asymmetric units [5,12,13] can be described as a “TBSV-like” (*Tombusvirus cucumber necrosis virus*) virus. This kind of particle could be seen in several plant viral capsid proteins: tobacco necrosis virus, noroviruses, and caliciviridae [14], and concluded with “Icosahedral viral capsid protein, S domain” (IPR000937) in InterPro. Therefore, the hypervariable P-domain affects the host specificity of each *Nodaviridae* genus. Alphanodvirus lacks P-domain. Betanodavirus and gammanodavirus share similar secondary structure similarity of the subunit capsid, each particle consists of a jelly-roll β-strand and protrusion domain. The P-domain contains hypervariable surface regions and specific DxD motifs, which contribute to the host specificity of these viruses [1,15]. The P-domain consists of a relatively highly disordered protruding area that possesses two anti-parallel β-sheets (67.9%) connected with two loop regions (32.1%) [16,17]. In addition, deletion of the last 26 amino acid residues of the MrNV346~371 capsid protein, which removes two protruding β-strands, reduces infectivity [18]. Several studies] have attempted to predict epitopes or hypervariable sites for betanodavirus vaccine development. Unfortunately, these epitopes are either too long (91–162, 195–214) [19,20] or too short (254–256; 223–227, 233–237, 253–259, 285–291), allowing selective identification of pathogenic segments. Moreover, P-domains of gammanodavirus represent a blade like structure which show a different form and mechanism to alphanodvirus and betanodavirus [21].

Immunologists have developed an integrated method for vaccine development based on the analysis of the protein sequences and the structures of target viruses [22]. For example, major capsid protein (MCP) can be assembled into virus-like particles (VLPs). B-cells play an important role in the immune system. Immunoglobulin, with the same antigen specificity, is secreted as an antibody by terminally differentiated B cells. Membrane-bound immunoglobulin on the surface of B cells serves as a receptor for antigens and is known as the B-cell receptor (BCR). Each of them is associated with unique receptor specificity. When a BCR binds to a cognate antigen, the B-cell receptor is stimulated to undergo proliferation. This involves the generation of plasma B cells and memory T cells [23]. Antigens are typically too large to bind to any receptor. Hence, partial antigen segments located on surface areas, called epitopes, are recognized by specific antibodies. Epitopes are generally divided into two categories: linear epitopes (LE), where a stretch of continuous amino acids is sufficient for binding, and conformational epitopes (CE), consisting of key amino acid residues that are brought together by protein folding [24]. CE prediction requires antigenic structures, mostly those of major capsid proteins (MCPs). MCP protein structure is also needed to further host-specific structural analysis to be resolved prior to conformational analysis. However, only a few protein structures have been resolved for the *Nodaviridae*. We applied both the empirical approaches (Phyre2 [25] and I-TASSER [26]) and state-of-the-art computational (RoseTTAFold) [27] methods. We focused on LE prediction, because information on the corresponding antigenic structures is scarce. The ideal predicted peptides should effectively elicit antibodies from specific hosts that recognize antigens and provide protection against infections [28]. Therefore, the peptides selected for vaccine design should ideally be conserved across different stages of the pathogen and possess binding affinity for the major populations of specific hosts. In this report, we present an integrated computational system incorporating a multi-expert voting mechanism and host-specific and surface structural analytics for LE prediction. Five existing epitope prediction tools including LBTOPE [29], BepiPred [30], BCPreds [23], ABCPred [31], and LEPS [32] were applied, and three important features were considered, including length constraint, physicochemical characteristics, and host-specific features. Candidates were screened based on these features from the five selected prediction systems, using different approaches and databases. The predicted epitopes were analyzed through surface structural characteristics and experimentally verified.

## 2. Materials and Methods

Host species and their corresponding major capsid protein sequences were retrieved from NCBI [33] and UniProt [34]. Due to the characteristics of *Nodaviridae*, the P-domain section was extracted from the full sequences of specific trials. In addition, we applied Phyre2, I-TASSER, and RoseTTAfold to predict the three-dimensional models, because the MCP structures of certain species in *Nodaviridae* have not yet been resolved. We retrieved the resolved structures of *Nodaviridae*.pdb files from RCSB. Species names were acquired from the ICTV taxonomy. In total, 3 genera and 18 species of *Nodaviridae* were collected. The genera, tentative species, and specific infected hosts are listed in Table 1.

### 2.1. Multi-Expert Voting Mechanism-Based LE Prediction

Our system is a metamodel that ensembles results from several existing prediction servers. In this case, we integrated five LE prediction tools into our voting mechanism to predict LEs for each representative antigenic sequence, including LBTOPE, BepiPred, BCPREDS, ABCPred, and LEPS. Each virus group was analyzed using five LE prediction tools designed to identify each antigenic residue as an epitope or a non-epitope residue. Each residue in a query antigen sequence possesses a corresponding score from the five different prediction tools. A higher score represented a higher possibility of the residue being an epitope. For comparison of the two subfamilies, if a continuous segment is predicted as an epitope segment within a subfamily, the system will check whether aligned segments from the other subfamily are also predicted as epitopes; if so, the system classifies this segment as a conserved epitope in both subfamilies, otherwise it is identified as a unique epitope. The observed variations in sequence or structural alignments could be associated with host specificity. The exclusive epitope segments within the same clustered antigen subfamily may play important roles in binding with antibodies present within the same group of host species. Multiple sequence alignments were performed using T-Coffee [35]. We also applied the protein structure prediction method, RoseTTAFold, to predict the P-domain of *betanodaviridae*. To make a difference between homogeneous and computational methods, we chose a P-domain section of resolved structure “4RFU” and a structure which lacked P-domain“3JBM” to predict protein structures.

### 2.2. Validation Method

In addition to the multi-expert voting mechanism prediction model, we proposed an additional complete sequence search (CSS) and a validation model consisting of a variety of propensity scales for enhanced evaluation. CSS applies BLAST tools (BLASTp-short) to search for all existing known and experimentally proven antigen peptides from the largest IEDB database. If experimentally proven epitopes in the IEDB could be matched by the predicted segments, there would be a higher possibility that the predicted LEs are genuine epitopes. Furthermore, we statistically analyzed the predicted LEs according to their residue contents and physicochemical properties for a reinforced classifier design. These features are introduced as follows.

Amino acid pairs (AAP) were generated by scanning the peptides using a window of two residue lengths and calculating the frequency of occurrence for each AAP. In total, 400 AAPs were generated. If a query peptide contains AAPs that belong to AAPs within true epitopes, there is a greater chance that the peptide could be considered an epitope. The SVMtrip_16AA [36] dataset contains two subsets: positive (LE) and negative (non-LE). We calculated the occurrence frequencies of AAPs between these two subsets to indicate the tendencies of epitopes and non-epitopes. Here, we refer to a previous study [37] to determine the frequency of occurrence of each pair. $f_{AAP\_i}^{+}$ and $f_{AAP\_i}^{-}$ are the occurrence frequencies of given *AAP_i* in the epitope and non-epitope set. $N_{AAP\_i}^{+}$ and $N_{AAP\_i}^{-}$ are the number of the specific *i*th AAP from 400 possible AAPs in the epitopes and non-epitopes. Finally, $Total_{AAP}^{+}$ and $Total_{AAP}^{-}$ are the total number of 400 AAPs in the epitope set and the non-epitope set. The differences between the two subsets can be interpreted as a likelihood ratio and normalized as an AAP antigenicity scale, $Norm(R_{AAP})$, by the Equation (1).
(1)$$f_{AAP\_i}^{+}=\frac{N_{AAP\_i}^{+}}{Total_{AAP}^{+}}\;,\;f_{AAP}^{-}=\frac{N_{AAP\_i}^{-}}{Total_{AAP}^{-}}\;,\;1\leq i\leq 400\;R_{AAP\_i}=\log\left(\frac{f_{AAP\_i}^{+}}{f_{AAP\_i}^{-}}\right)\;Norm(R_{AAP\_i})=\left(\frac{R_{AAP\_i}-Min(R_{AAP})\;}{Max(R_{AAP})-Min(R_{AAP})}\right)$$

The position-specific scoring matrix (PSSM) is a commonly used representation of sequence motifs. PSSM is a position weight matrix (PWM) that can distinguish evolved sequences and genuine binding sites among similar sequences [38]. First, a position frequency matrix (PFM) creates a column for each amino acid, corresponding to a total of 20 rows for amino acids in the protein sequences. An alignment result *X* is given. Each column is created by calculating the occurrence of each position in a sequence. Second, a position probability matrix (PPM) can be created by dividing the occurrence counts by the number of sequences. If we give a set of *N*-aligned sequences for the sequence length of *L* residues: alignment result *X* (size *L*N*), the value in a corresponding PPM (matrix *M*) can be calculated using Equation (2). *I* (*X_i,j_* = *k*) is an indicator function, where *I* (*a = k*) is 1 if *a = k* and 0 otherwise. Additionally, we applied a variable *b* = 1/*k* as a background model (expected probability), where *k* is the total number of amino acids (i.e., *k* = 20). PPM can be converted to PWM using Equation (3). We can convert a sequence to parameter information content (IC) using amino acids with corresponding positions in the PWM.
(2)$$M_{k,j}=\frac{1}{N}\overset{N}{\underset{i=1}{\sum }}I\left(X_{i,j}=k\right);\;\;i\in \left(1,\ldots ,N\right),\;j\in \left(1,\ldots ,L\right),\;1\leq k\leq 20$$
(3)$$M_{\mathrm{PWM}}={\;\log}_{2}\left(\frac{M}{b}\right)$$

For the propensity scale, sequences were scanned using a sliding window constituting the central residues *i* and neighborhood residues (*i* ± ½* window size). We assigned a value of 7 to the window size parameter [39]. We also applied four physicochemical scales extracted from the ProtParam tool for reinforced evaluation [40], including hydrophobicity [41], flexibility [42], surface accessibility [43], and polarizability [44]. In addition, we adopted the “surface patch” strategy to describe the local spatial context of each residue in the predicted epitopes. Commonly, the surface patch consists of some spatially adjacent surface residues and the central residue itself and is classified as epitope patch and non-epitope patch according to the state assigned to the central residues [45]. In this study, the surface patch consisted of the residues of predicted epitopes, and the middle residue of the peptide sequence was taken as the central residue. To gain insight into the common structural contents or physicochemical characteristics of the predicted LE, the surface patch was evaluated based on the presence of several known features. To measure the spatial features of the adjacent residues, we considered whether the distance between adjacent residues and the central residue could impact antigenicity and calculated the average distance of the surface patches. Furthermore, the contributions of interior and surface residues were also taken into consideration. If the relative accessible surface area (RASA) calculated by the DSSP [46] program was greater than 5%, the residue was considered to be a “surface residue” [47]. Subsequently, the ratio of the number of surface residues to the number of residues in the interior of the peptide was calculated. Non-polar molecules exposed to water are unfavorable and hydrophobic molecules are usually located in the center, therefore hydrophilic and surface-exposed amino acids are preferable. The values of half-sphere exposure (HSE), which is widely used in protein structural analysis and provides relatively more geometric information than other measurements, were also calculated in this study. A larger HSE value indicates that the Cα of the central residue is more adjacent to other Cα atoms [48]. Finally, the residue depth from a Cα atom to the protein surface for query residues was also considered. Both the HSE and the residue depth were obtained using MSMS [49] and Biopython’s Bio.PDB package. In this phase, a support vector machine (SVM) classifier with “RBF” kernel was applied to train the prediction model. B cell epitope datasets were taken from SVMTrip_20AA [36], Chen’s database, Epitopia [50], and Bepipred-2.0 [30]. Among them, a total of 6969 epitopes and 6962 random peptides were collected. Datasets for evaluation were obtained from IEDB and Uniport, which contain 20,335 experimentally validated epitopes and 20,161 randomly selected peptides. We also calculated two properties of surface shape—shape index and curvedness—to evaluate the predicted epitopes from resolved protein structures. The shape index (*S_i_*) is a number ranging from –1 to 1, the larger number the number of curvedness the more it shows how curved the object is, which describes the shape of the local surface at any given point and is independent of the scale of the surface. Points with positive values represent convex doom and negative values represent concave cup. We converted epitope patches from .pdb file to point cloud format with MSMS and applied pymesh [51] to calculate mean and Gaussian curvature to each vertex of the model. Shape index and curvedness were calculated by Equation (4) and Equation (5), respectively. For any vertex on a surface, there will be two points at which the curvature reaches a maximum *K_max_* and minimum *K_min_* [52,53].
(4)$$S_{i}=\frac{-2}{\pi }\arctan\left(\frac{K_{max}+K_{min}}{K_{max}-K_{min}}\right)$$
(5)$$\;\mathrm{Curverdness}=\sqrt{\frac{K_{max}{}^{2}+K_{min}{}^{2}}{2}\;}$$

### 2.3. Biological Assays

According to the constructed prediction systems and in silico validation principles, the predicted exclusive LEs and reference segments for betanodaviruses were chemically synthesized by PepPower™ Peptide Synthesis Technology (GenScript, Piscataway, NJ, USA). After synthesis, the synthesized epitopes were used as antigens for antigenicity tests. All peptide samples were proceeded in triplicate by immunoassays. The assays were performed using an enzyme-linked immunosorbent assay (ELISA) to validate the antigenicity of the predicted LEs. This immunological analysis is very sensitive and highly specific for the detection and quantification of substances such as antibodies, antigens, and other proteins. The antigen-containing samples were coated on 96-well microplates containing polystyrene and incubated until they were adsorbed onto the surface of the microplate with coating buffer (0.2 M sodium carbonate/bicarbonate, pH 9.4). The coating buffer immobilizes antigens, which leads to maximal adsorption on the microplate surface and optimization of interactions with the detection antibody. The hydrophobic sites were exposed after the antigens were adsorbed onto the microplates. The blocking processes were used to fill the interspaces with bovine serum albumin (BSA), non-fat milk powder, or casein to block nonspecific binding. Washing steps were required to eliminate unbound and excessive components that might interfere with the assay. First, 10 µg of antigens (synthesized peptides) were applied to a 96-well microplate, incubated for 1 hat RT, and blocked with wash buffer (PBST buffer, 1× phosphate-buffered saline with 0.1% Tween 20) to remove non-specific antigens. Next, the cells were treated with the rabbit pre-immune antibody and rabbit post-immune antibody (rabbit anti-NNV capsid protein antibody) and were incubated for an hour and washed 3–5 times. Finally, hybridization with a secondary antibody (goat anti-rabbit IgG (H+L) antibody) conjugated with alkaline phosphatase (AP, Jackson ImmunoResearch, West Grove, PA, USA) was used as the detection antibody. After hybridization, the microplate was also washed 3–5 times to remove non-specific antibodies with wash buffer, and substrate pNPP (para-nitrophenylphosphate, ThermoFisher) was added for AP detection and read at 405 nm using an ELISA reader. The ELISA results were further compared and analyzed before and after immunization using GraphPad Prism (version 5.0; GraphPad Software, Inc.). The ELISA results were further statistically analyzed by t-test for each synthesized peptide (*n* = 3, *p* < 0.05).

## 3. Results

### 3.1. Prediction Results between Alphanodavirus and Betanodavirus

There was no conserved LE between alphanodavirus and betanodavirus clusters, but eight exclusive LEs were found for the alphanodavirus subfamily and two exclusive LEs for betanodavirus. The sequence similarity between alphanodavirus (subunit particles, PDB:1NOV) and betanodavirus (S-domain; PDB 4RFT) is 22.67%. None of the predicted LEs for alphanodavirus could be found in the experimentally verified database IEDB, while the two predicted LEs for betanodavirus could be matched with existing reports from IEDB. In addition, a previous study reported and validated the true epitope segment of “BFNNV_261~272_:RPLSIDYSLGTGD” using biological experiments [19]. Through in silico scanning of IEDB, our prediction system increases the opportunity to predict genuine epitopes.

### 3.2. Prediction Results between Betanodavirus and Gammanodavirus

In the second trial, the predicted LEs located within the P-domain of betanodavirus and gammanodavirus were compared. In addition, P-domain segments from the grouper-infecting betanodavirus subfamily were exclusively predicted for detail comparison. The sequence similarity of the P-domain of betanodavirus (PDB:4RFU) and gammanodavirus (PDB:5YKV) was 27.18%, and the root-mean-square error (RMSD) between the two structures was 3.418 Å. No conserved epitopes were found in these two clusters. In contrast, four unique LEs in the betanodavirus group and two unique LEs in the gammanodavirus group were detected. The results of the four predicted LEs for grouper-infecting betanodavirus are shown in Table 2. Most of the predicted LEs were exactly or partially identical to the predicted LEs of the whole betanodavirus subfamily. For example, BFNNV_283~295_:KKVAGNVGTPAGW was also predicted in the grouper-infecting betanodavirus subfamily trial (DGNNV_283~295_:KKFAGNAGTPAGW); BFNNV_302~322_:DNFNKTFTQGVAYYSDAQPRQ in the betanodavirus trial was split into BFNNV_302~309_:DNFNKTFT and BFNNV_314~322_:AYYSDAQPRQ in grouper-infecting virus trials (DGNNV_302~309_:DNFNKTFT; DGNNV_314~322_:AYYSDEQPRQ).

In Figure 1a, we selected one subunit (PDB:4RFU) and one chain of this subunit to visualize the predicted epitopes with cartoon style. Each epitope was represented by distinct color codes. Identical or partially duplicated epitopes in betanodavirus and grouper-infecting betanodavirus trials were colored the same. In Figure 1b, we also marked the residues with the following color codes: yellow (highly variable regions), pink (positive-charged residues), blue (negative-charged residues). In summary, the predicted epitopes for betanodavirus were charged residues and well conserved within highly variable regions.

There were four predicted LE epitopes for betanodavirus; among them, three segments were considered epitope candidates and were synthetized for biological experiments. BFNNV244–251:GSTQLDIA is the only unique epitope in the betanodavirus. We synthesized the predicted LEs of betanodavirus that overlapped among the different trials (Table 2). Through comparison, we selected exclusive epitopes for betanodavirus (BFNNV_244–251_:GSTQLDIA, BFNNV_261~272_:RPLSIDYSLGTGDV, BFNNV_283~295_:KKVAGNVGTPAGW, and BFNNV_300~321_:LWDNFNKTFTQGVAYYSDAQP) and grouper-infecting betanodavirus (DGNNV_221–238_:PIMTQGSLYNDSLSTNDF and BFNNV221–238:PILTLGPLYNDSLAANDF). To compare with previous research results, we synthesized EFNNV/GNNV_249–258_:DIAPDGAVFQ as a reference for antigenicity comparison with our predicted LEs [54]. An ELISA was performed to identify the host specificity of NNV for grouper species. The results revealed that a significant change occurred before and after immunization. These predicted LEs reflect a strong antigenic response in grouper species. In Figure 2, enzyme-linked immunosorbent assays (ELISAs) were performed to identify host-specific LEs. Synthetic peptides (10 μg) or coating buffer were coated on a 96-well microplate. All peptides were labeled with primary antibody (rabbit anti-NNV coat protein antibody) and secondary antibody (goat anti-rabbit IgG (H+L) conjugated alkaline phosphatase. Detection was performed at 405 nm after adding the alkaline phosphate substrate. The x-axis represents the comparison of antibodies against NNV before and after immunization. The y-axis indicates the absorbance value at 450 nm. The responses of the synthetic peptides for betanodavirus (BFNNV_CP244–251, 261–272, 283–295, and 302–322) are shown from (a–d). The response of the synthetic peptide for grouper-infecting betanodavirus (DGNNV_CP221–238) is shown in (e). The reference control EFNNV_CP249–258 was based on a previous report and is shown in (f). The comparison of pre-immunization and after immunization, the ELISA results revealed each candidate peptide with antigenicity after immunization in rabbit anti-NNV CP antibody showing significant differences (*n* = 3, *p* < 0.05) (Appendix A).

In Figure 3a, we applied traditional empirical approaches (Phyre^2^ and I-TASSER) and state-of-the-art computational (RoseTTAFold) methods to predict the P-domain Betanodavirus “4RFU”and “3JBM”. The RMSD of each predicted model between are: Phyre^2^ 0.3; I-TASSER 0.5; RoseTTAFold 1.07. It is surprising that the traditional homology method achieved a lower RMSD for a better prediction. In Figure 3b, the predicted epitopes are shown by structurally aligning with the resolved structure “4RFU” and RoseTTAFold predicted “3JBM”. Only 3JBM_49~65_:LSIDYSLGTGDV/4RFU_49~65_:LSIDYSLGTGDV and 3JBM_117~201_:VCTRVO/4RFU117~201:VCTRDSX show a difference in substructure, and the mapped epitopes do not show obvious differences.

For gammanodavirus subfamily analysis, two conserved epitopes including MrNV_257~276_:YNADTIGNWVPPTELKQTYT and MrNV353~360:AVDPKPYQ were colored with both cartoon-style and space-filled 3D structured models and shown in Figure 4a. Figure 4b shows the aligned results by the self-developed multiple structure alignment system(AIR system). The charged residues and hypervariable regions of gammanodavirus and the sequence or structurally aligned results of PvNV (PDB:5YKZ) and MrNV (PDB:5YKV) were shown. The RMSD of these two structures was 1.607 Å, despite their relatively low sequence similarity. It is worth noting that PvNV353~360:ASKKQTTG and MrNV353~360: AVDPKPYQ were located in regions of high variability and exhibited symmetrical structures in three dimensions. The β-strand peptide located at the C-terminus of the P-domain containing the last 26 amino acids (MrNV346~371:LVTDYQGAVDPKPYQYRIIRAIVGNN) were related to infectivity. MrNV353~360:AVDPKPYQ and DGNNV261~272:RPLSIDYSLGTGDV from betanodavirus showed similar properties (β-strand, charged, mostly protrusion shaped).

In Figure 5, two types of NNV P-domain sequences performed by structural and sequence alignments (PDB:4RFU and 5YKV) with secondary structure annotations are shown. It can be observed that despite the P-domain of two structures they share low similarity (RMSD:3.418) but show close homology with similar compositions of secondary structures (β-strands).

### 3.3. Prediction Results between Alphanodavirus and Gammanodavirus

In the third trial, one conserved LE, five unique LEs for the alphanodavirus, and two unique LEs for gammanodavirus were identified. The sequence similarity between alphanodavirus (subunit particles; PDB:1NOV) and gammanodavirus (S-domain; PDB:6AB6) was 21.28%. Five unique LEs for alphanodaviruses were Nov_67–81_:DFSTDPGKGIPDKFQ, Nov_137–147_:PGFDQLFGTSAT, Nov_176–186_:AGSIQVYKIPL, Nov_258~268_:PPANVTNAQAS, and Nov_325~338_:ARESPANDEYALAA, and two unique LEs for gammanodavirus were MrNV_41~49_:VAKPTVAP and MrNV_110~135_:SQFWERYRWHKAAVRYVPAVPNTLAC. Alphanodavirus lacks the P-domain, therefore the whole subunit structure and predicted epitopes are visualized and colored in Figure 6. As shown in Figure 6a, the NNV S-domains of three subfamilies containing alphanodavirus (1NOV), betanodavirus (4RFT), and gammanodavirus (6AB6) were determined by sequence alignment with secondary structure annotations from DSSP. It can be observed that the core structure of the S-domain is mainly composed of beta-turn elements and possesses a composition similar to the secondary and tertiary structures. Therefore, the S-domain of NNV was relatively stable compared with the P-domain during evolution. The subunit (PDB:1NOV), trimer particle model, and total predicted epitopes are shown in Figure 6b. In Figure 6c, three types of NNV (subunit particles PDB:1NOV 6AB6; S-domain: 4RFT) were structurally aligned. The RMSD between the three structures was 2.761 and four types of alphanodaviruses (Pav:1F8V, BBV:2BBV, FHV:4RFT, and Nov:1NOV) were structurally aligned. The RMSD between the four alphanodavirus S-domain structures is 1.195. Compared with both gammanodaviruses and betanodaviruses, alphanodaviruses possess two unique protrusion structural segments (NOV_187~225_:KQVLNSYSQTVATVPPTNLAQNTIAIDGLEALDALPNNN and Nov_258~281_:PPANVTNAQASMFTNLTFSGARYT), which are well conserved in the alphanodavirus genus. It was also observed that the predicted epitopes NOV_200~226_:VPPTNLAQNTIAIDGLEALDALPNNNY and NOV_258~281_:PPANVTNAQAS were located at the unique protrusion structure shown in Figure 6c.

### 3.4. Physicochemical Characteristics of Predicted Residues

The amino acid index was obtained by calculating the occurrence frequency, which is defined as the number of predicted epitope residues divided by the overall residues and surface residues. Glycine (G) and alanine (A) accounted for the two highest ratios of predicted epitopes. In contrast, histidine (H) and glutamine (Q) are less likely to be considered epitope residues. The charge states of the predicted residues were further examined, as shown in Figure 1, Figure 3 and Figure 5. Most of the peptides were located in a highly variable state and contained positively or negatively charged residues. Furthermore, the peptides were transformed into a point-cloud data structure and their corresponding mean curvature, Gaussian curvature, and shape index were analyzed, as shown in Figure 7. Most shape indices of the predicted epitopes were −1–−0.75 (spherical cup ~ rut) as a receptor.

## 4. Discussion

A comprehensive LE prediction system for host-specific antigens has been proposed. For group feature detection, the antigen sequences were clustered prior to importing the sequences into the proposed system. For example, the *Nodaviridae* family can be categorized into three different subfamilies: alphanodvirus, betanodavirus, and gammanodavirus. In this study, we applied different combinations of existing resources to predict the conserved and unique LEs. Antigen sequences of each subfamily were analyzed using a multi-expert voting mechanism, and multiple structural alignments were performed to confirm the conserved and unique characteristics. Using multiple sequence aligned locations, the consensus voting module selected epitope candidate residues by accumulating votes provided by five different renowned LE epitope prediction tools. In addition to individually voted epitope residues, the minimum lengths of concatenated epitope residues were required for further experimental design. All LE candidates for different host-specific groups were cross identified. In addition, to increase the success rates of the eventual vaccine design, we emphasized the surface structural characteristics of the predicted epitopes. Therefore, antigens without resolved structures can be analyzed by applying structure prediction systems to their corresponding virtual structure predictions. The predicted epitope residues were reconfirmed for their surface conditions and aligned using structural alignment tools for revalidation. Based on the alignment results, the predicted conserved and/or unique LEs for different subfamilies were sequentially and structurally distinguished. The proposed system was compared with existing resources by selecting a well-known prediction system for comparison. We selected the ABCPreds prediction system and applied the betanodavirus antigen sequence as a test case. In this case, ABCPreds predicted 10 epitopes, one of the betanodavirus sequences, including 86% of the query antigen sequences. Conversely, our system predicted only four significant epitopes, and the total length of the predicted epitopes was 38.7% of the query antigen sequence. A unique aspect of the proposed system is that in addition to achieving accurate prediction, it provides host-specific LE prediction as long as the antigen sequences can be clustered in advance. Through biological experiments, three of the predicted epitopes were initially validated as accurate epitopes with strong antigenicity responses. Our system utilizes host-specific features to predict effective epitopes for biologists, and the developed multi-expert voting mechanism-based LE prediction system can successfully predict LEs with significant antigenic specificity. In the antigenicity assay, we found that there were many overlapping LEs in the betanodavirus subfamily. To further compare the differences among these predicted LEs, five predicted LEs were selected, synthesized, and analyzed by ELISA. In a recent report, an epitope of EFNNV/GNNV_249–258_:DIAPDGAVFQ was shown to be effective in the giant grouper (*Epinephelus lanceolatus*); therefore, this epitope was selected as a reference for antigenicity analysis. As revealed by the ELISA tests, all predicted LEs displayed high antigenicity for the orange spot grouper. Interestingly, the last three amino acids in the predicted LE BFNNV_244~251_:GSTQL***DIA*** were the first three residues in EF/GNNV_249~258_:***DIA***PDGAVFQ. This suggests that they might play an important role in grouper immunity. The predicted LEs were approximately 8–22 residues in length; the majority of the residues were located at the N-terminus of the capsid protein and characterized by adequate antigenic properties and host specificity. Therefore, we hypothesized that the predicted epitopes were involved in the adaptive immunity of groupers. However, further investigation via in vivo analysis is required to confirm this hypothesis. In conclusion, this prediction system based on host-specific characteristics provides important and exclusive information to fish immunologists for developing fish vaccines in an effective and efficient manner.

## Figures and Tables

**Figure 1 viruses-14-01357-f001:**
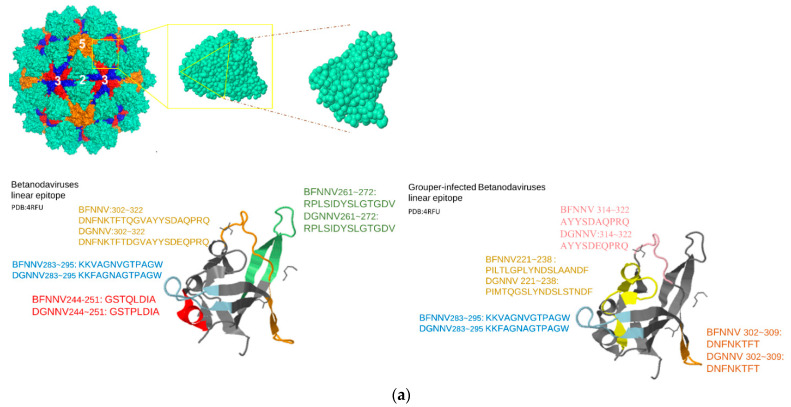
Predicted LEs for betanodavirus and grouper-infecting betanodavirus. P-domain sequence alignment with variable and charged annotation. (**a**) Visualize the predicted epitopes with cartoon style. Predicted epitopes were shown in different colors; (**b**) sequence alignment result and the corresponding hypervariable and charged residues.

**Figure 2 viruses-14-01357-f002:**
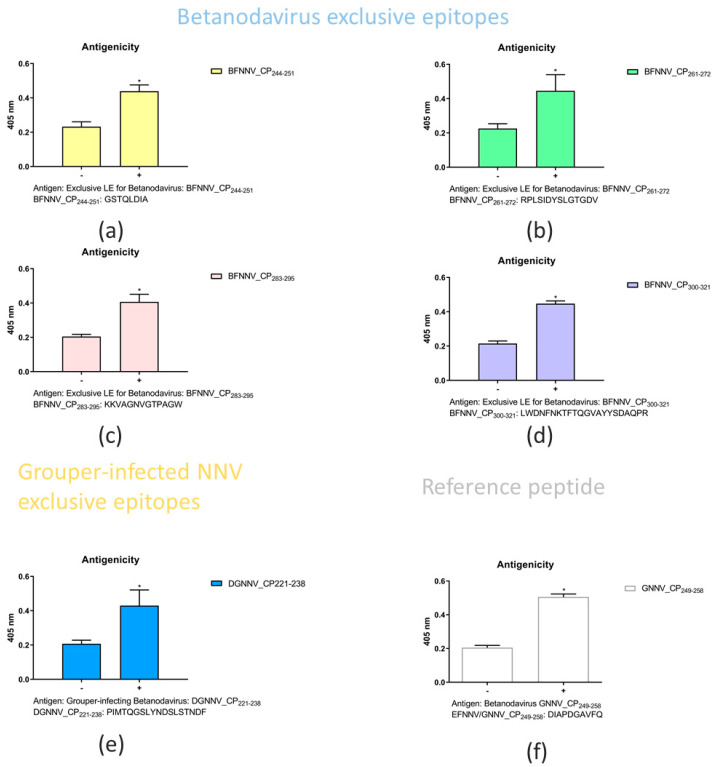
Enzyme-linked immunosorbent assays (ELISAs) were performed to identify host-specific LEs. From (**a**–**d**), 4 betanodavirus exclusive epitopes; (**e**) grouper-infecting virus exclusive epitope; (**f**) a reference peptide. The symbol “-” represents the case of pre-immunization (without immunization), and “+” represents after immunization (rabbit antibody was immunized by NNV capsid protein). All ELISA values were further analyzed by GraphPad Prism 8.0 software with t-test (*n* = 3, * *p* < 0.05). Detailed experimental data can be found in Appendix A.

**Figure 3 viruses-14-01357-f003:**
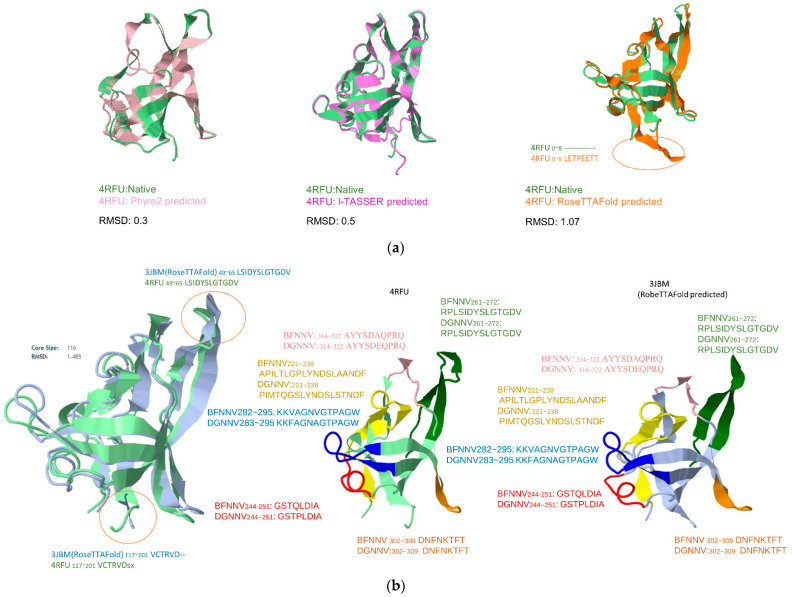
Structural alignment of predicted structures: (**a**) the resolved structure of RSCB PDB: 4RFU was aligned with 3 predicted structures by Phyre2, I-TASSER, and RoseTTAFold, respectively; (**b**) structural alignment and predicted epitopes of resolved 4RFU and the predicted 3JBM model by RoseTTAFold. Marked circles represent segments with major difference between predicted and resolved structures.

**Figure 4 viruses-14-01357-f004:**
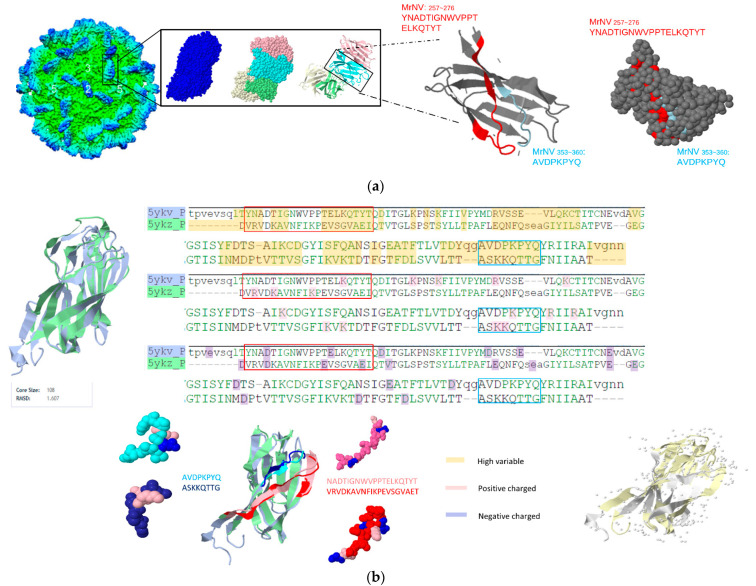
Predicted LEs for gammanodavirus and alignment between MrNV(PDB:5YKV) and PvNV(PDB:5YKZ). (**a**) Visualizing the predicted epitopes with space fill and cartoon style, each chain in a subunit and each epitope in a chain was represented by distinct color codes; (**b**) the P-domain of MrNV (PDB:5YKV) and PvNV (PDB:5YKZ) are structurally aligned by our own developed structural alignment system (AIR), represented by the following color codes: yellow (highly variable region), pink (positive-charged residue), blue (negative-charged residue).

**Figure 5 viruses-14-01357-f005:**
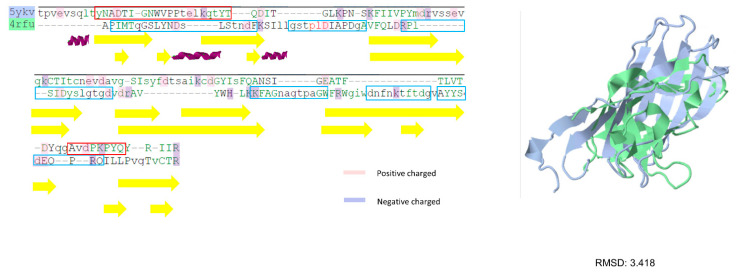
Sequence alignment between betanodavirus (PDB:4RFU) and gammanodavirus (PDB:5YKV). P-domain of betanodavirus and gammanodavirus are sequentially and structurally aligned. The alignment is annotated by secondary structures (helix, turn, and strand) acquired from DSSP.

**Figure 6 viruses-14-01357-f006:**
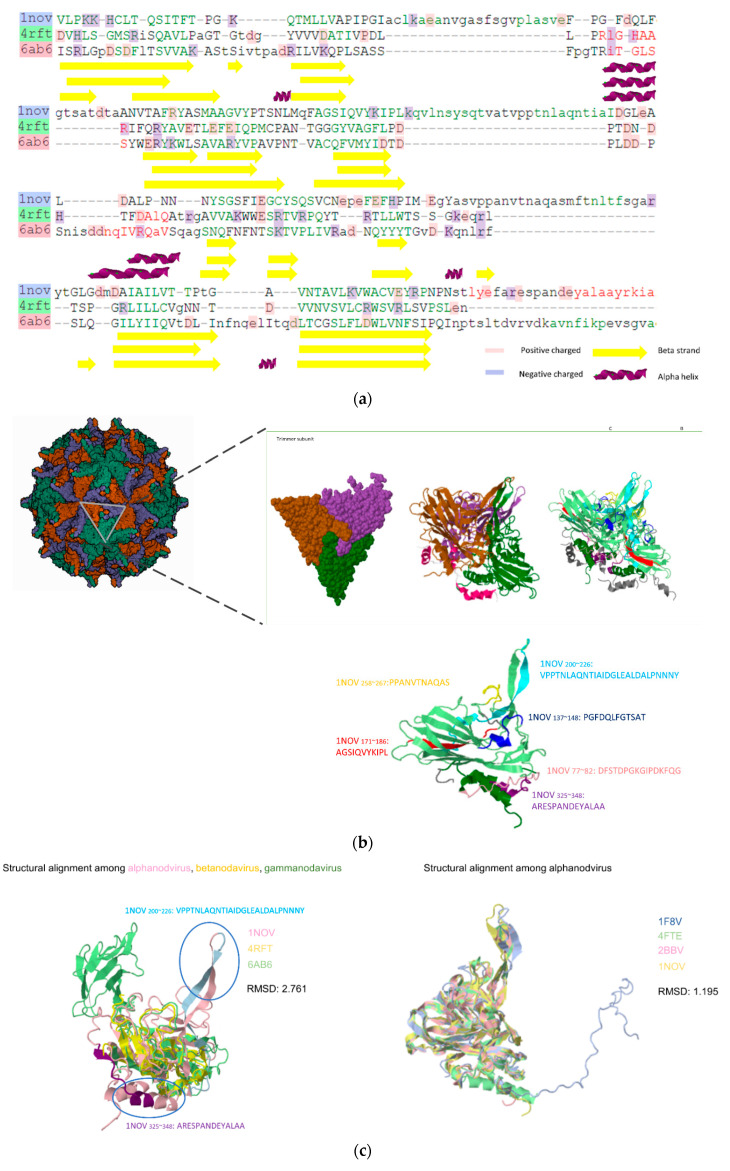
Summary of the icosahedral structure of alphanodavirus containing 180 subunits of the capsid protein. We chose a tangled subunit (PDB:1NOV) and a chain of this subunit to visualize the predicted epitopes using the spacefill and cartoon representation. (**a**) Sequence alignment of NNV S-domain in (subunit particles PDB:1NOV 6AB6; S-domain: 4RFT). (**b**) A subunit particle of alphanodavirus (PDB:1NOV) and a trimer P-domain model with ribbon and spacefill representation. Four predicted epitopes are represented using distinct color code from Figure 5a. (**c**) Two structure alignment trials of S-domain among: (left) alphanodavirus (PDB:1NOV), betanodavirus (PDB:4RFT), and gammanodavirus (PDB:6AB6); (right) alphanodavirus subfamily (PDB:1F8V, 4TFE, 2BBV, and 1NOV). Unique protrusion structure and part of predicted epitopes are labeled.

**Figure 7 viruses-14-01357-f007:**
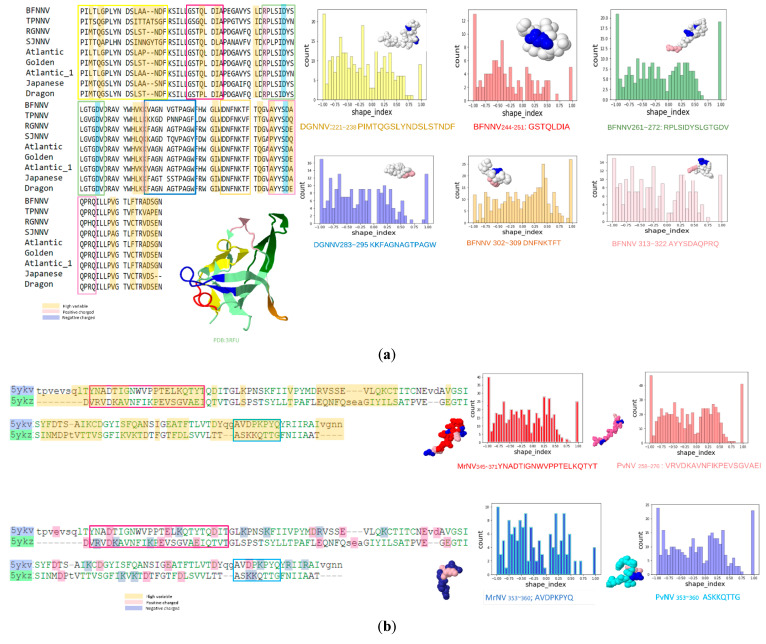
Shape index of predicted epitope from betanodavirus and gammanodavirus. The surfaces of the predicted epitopes were extracted from betanodavirus (4RFT) and gammanodavirus (5YKV and 5YKZ) with MSMS. PyMesh is then applied to calculate the Gaussian and mean curvatures for each vertex and to further calculate the shape index (SI). Each histogram is represented using the corresponding color of the peptide from the sequence alignment. (**a**) Epitopes of betanodaviruses; (**b**) epitopes of gammanodavirus.

**Table 1 viruses-14-01357-t001:** Selected genera of the *Nodaviridae* family and infected host species.

Genus	Selected Species	Hosts	RCSB ID	NCBI GenBank
Alphanodviruses	*Flock House virus* (FHV)	Barley/Saccharomyces Cerevisiae/Moth/Beetle	4FSJ 6ITB 6ITF 4RFT	EF690538.1
	*Black Beetle virus* (BBV)	Beetle	2BBV	X00956.1
	*Drosophila melanogaster American nodavirus* (DmANV)	Bee		GQ342966.1
	*Nodamura virus* (Nov)	Moth/Mosquito/Bee Wild swine	1NOV	NC_002691.1
	*Boolarra virus* (Bov)	Moth		NC_004145.1
	*Pariacoto virus* (Pav)	Moth	1F8V	NC_003692.1
Betanodaviruses	*Striped jack nervous necrosis virus* (SJNNV)	Bass		NC_003449.1
	*Tiger puffer nervous necrosis virus* (TPNNV)	Puffer		NC_013461.1
	*Atlantic halibut nodavirus* (AHNV)	Halibut		AY962682.1
	*Golden pompano nervous necrosis virus* (GPNNV)	Pompano		HQ859934.1
	*Atlantic cod nodavirus* (ACNV)	Cod		ABU95413.1
	*Japanese flounder nervous necrosis virus* (JFNNV)	Flounder		BAB00609.2
	*Dragon grouper nervous necrosis virus* (DGNNV)	Grouper	3JBM	AAG22496.1
	*Barfin flounder nervous necrosis virus* (BFNNV)	Flounder		NC_013459.1
	*Redspotted grouper nervous necrosis virus* (RGNNV)	Grouper	3JBM	NC_008041.1
	*Epinephelus coioides nervous necrosis virus* (GNNV/EFNNV)	Grouper	4RFU (P-domain only) 4RFT (S-domain only) 4WIZ	MG874758.1
Gammanodaviruses	*Macrobrachium rosenbergii nodavirus* (MrNV)	Macrobrachium rosenbergii	6H2B 6JJC 5ykv	NC_005095.1
	*Penaeus vannamei nodavirus* (PvNV)	Whiteleg Shrimp	5YKZ (P-domain only) 5YL0	NC_014977.1

**Table 2 viruses-14-01357-t002:** Predicted LEs of grouper-infecting betanodavirus.

Nodaviridae	Predictive LEs of Representative Peptide	Residue Location	IEDB(CSS)	SVM_Classifier
Grouper-infecting betanodavirus	PILTLGPLYNDSLAANDF PIMTQGSLYNDSLSTNDF	BFNNV: 221~238 DGNNV: 221~238	136550	Y
	KKVAGNVGTPAGW KKFAGNAGTPAGW	BFNNV: 283~295 DGNNV: 283~295	N/A	Y
	DNFNKTFT DNFNKTFT	BFNNV: 302~309 DGNNV: 302~309	N/A	Y
	AYYSDAQPRQ AYYSDEQPRQ	BFNNV: 313~322 DGNNV: 313~322	N/A	Y

N/A represents the vacuity of similar experiment-proved epitopes in IEDB and segments with grey background represent matched peptide between IEDB’s assay and the proposed system.

## Data Availability

Not applicable.

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
