# Peer review of "Comprehensive Linear Epitope Prediction System for Host Specificity in Nodaviridae"

_viruses, 2022, doi:10.3390/v14071357_

Round 1
Reviewer 1 Report
The manuscript presents a method for predicting linear epitopes (LE) based on an integrated computational system that incorporates a multiple expert voting mechanism and surface and host-specific structural analyses. Despite the multiple image analysis and the fact that some peptides are shown to be immunogenic, some statistical analyses are necessary.
These are the main issues:
- It is neccesary to perform statistical analisys of antigenicity values shown in Fig 2. Are differences statistical significant?
- Fig 2. Replace benodavirus by betanodavirus in the Fig.2 and throughout the text.
- Please explain how rabbit were immunized with NNV and wich strain was used.
- Line 393. Change "enzyme-lynked" by "Enzyme-linked".
- Figure 6a. Please improve the quality of the image.
Reviewer 2 Report
Nodaviridae infection is one of the leading causes of death in commercial fishes. Although many vaccines against this virus family have been developed, their efficacies are relatively low. In the present study, the predicted epitopes were analyzed through surface structural characteristics and experimentally verified.In general, this is a very interesting subject. However, the manuscript should be much improved in logical organization before it is considered for publication.
(1) There are many errors in grammar and syntax throughout the text of the manuscript, which are required for correction.
(2) Line 82-87, Some virus names can be abbreviated.
(3) Line 342-343, In Fig (1b), We also marked the residues with the following color codes: yellow (highly variable regions), pink (positive- charge residues), blue (negative-charged residues).The first letter of We needs to be lowercase.
(4) Line 389-391, different font sizes.
(5) Line 393, Figure 2. enzyme-linked immunosorbent assays (ELISAs) were performed to identify host-specific LEs. The first letter of enzyme needs to be capitalized.
(6) Figure 6(a) and Figure7 are not clear.
(7) You compared the difference between betanodavirus and gammanodavirus, alphanodavirus and betanodavirus, alphanodavirus and gammanodavirus , so is there a difference between the three?
Reviewer 3 Report
Graphs in figure 2 need statistical significance details. The readability of the peptide sequences is very low resolution (same with figure 6a and 7)
Round 2
Reviewer 2 Report
The manuscript meets publication requirements.